



# Acceleration of protons and heavy ions to suprathermal energies during dipolarizations in the near-Earth magnetotail

Andrei Yu. Malykhin[1,2], Elena E. Grigorenko[1,2,3], Elena A. Kronberg[4,5], Patrick W. Daly[4], Ludmila V. Kozak[6,7]

[1]Space Research Institute of Russian Academy of Sciences, Moscow, Russia
[2]St. Petersburg State University, Saint Petersburg, Russia
[3]Department of Space Physics, Moscow Institute of Physics and Technology, Moscow, Russia
[4]Max Planck Institute for Solar System Research, Göttingen, Germany
[5]Ludwig Maximilian University of Munich, Munich, Germany
[6]Kyiv Taras Shevchenko University, Kyiv, Ukraine
[7]Space Research Institute National Academy of Sciences of Ukraine and State Space Agency of Ukraine, Kyiv, Ukraine

*Correspondence to*: *Andrei Yu. Malykhin* (amaurdreg@gmail.com)

**Abstract.** In this work we present an analysis of the dynamics of suprathermal ions of different masses ($H^+$, $He^+$, $O^+$) during prolonged dipolarizations in the near-Earth magnetotail ($X > -17$ $R_E$) according to Cluster/RAPID observations in 2001-2005. All dipolarizations from our data base were associated with fast flow braking and consisted of multiple dipolarization fronts (DFs). We found statistically that fluxes of suprathermal ions started to increase ~ 1 min before the dipolarization onset and continued to grow during ~ 1 min after the onset. The start of flux growth coincided with the beginning of decrease in the spectral index $\gamma$. The decrease in $\gamma$ was observed for protons during ~ 1 min after the dipolarization onset, and for $He^+$ and $O^+$ ions - during ~3 min and ~ 5 min after the onset respectively. The negative variations of $\gamma$ for $O^+$ ions were in ~2.5 times larger than for light ions. This demonstrates more efficient acceleration for heavy ions. The strong negative variations of $\gamma$ were observed in finite energy ranges for all ion components. This indicates the possibility of non-adiabatic resonant acceleration of ions in the course of their interaction with multiple DFs during dipolarizations. Our analysis showed that some fraction of light ions can be accelerated up to energies ≥600 keV and some fraction of oxygen ions can be accelerated up to ~1.2 MeV. Such strong energy gains cannot be explained by acceleration at a single propagating DF, and suggest the possibility of multistage ion acceleration in the course of their interaction with multiple DFs during the prolonged dipolarizations.

## 1 Introduction

One of the important processes in the dynamics of the Earth's magnetotail is magnetic dipolarization. This manifests itself in enhancement of the northward magnetic field component ($B_Z$), which results in transformation of the initially stretched magnetic configuration into the more dipole-like one. This process is often associated with an increase in geomagnetic activity (Sergeev et al., 2012 and references therein).

Spacecraft observations have shown that dipolarization phenomena can be divided into two main groups. The first group includes isolated dipolarization fronts (DFs) propagating along with the Bursty Bulk Flows (BBFs) towards the Earth (e.g. Angelopoulos et al., 1992; Nakamura et al., 2002; Runov et al., 2009). The DFs are usually observed during a few minutes or less (e.g. Schmid et al., 2011), and it is believed that they are formed downtail in the course of reconnection (e.g. Sitnov et al., 2009). The second group includes the so-called "secondary" dipolarizations related to the braking of fast flows and magnetic flux pile up in the near-Earth tail (e.g., Nakamura et al., 2009). The origin of "secondary" dipolarizations is still debated. They can be the consequence of magnetic flux pileup due to arrival of multiple BBFs (e.g. Liu et al., 2013; 2014) or they can be caused by the development of cross-tail current instability in the near-Earth CS (e.g. Lui et al. 2011). Usually the secondary dipolarizations are associated with the formation of the substorm current wedge (SCW) and are observed during up to several hours (McPherron et al., 1973; Sergeev et al., 2012).



The prolonged "secondary" dipolarizations have a complicated temporal and spatial structure. (e.g., Nakamura et al., 2009; Grigorenko et al., 2016; 2018, Malykhin et al., 2018a). They consist of the prolonged growth of the $B_Z$ field (during ~tens of minutes) along with multiple short (~1-2 min) $B_Z$ pulses. It was shown that strong enhancements of the dawn-dusk electric field ($E_Y$) are often observed along with the $B_Z$ pulses (e.g. Runov et al., 2011, Grigorenko et al., 2018).

Thus, it is natural to assume that such magnetic structures affect the dynamics and acceleration of charged particles. The processes of acceleration of charged particles at single DFs have been studied in detail by using both spacecraft observations and kinetic simulations. Fu et al., (2011) using Cluster observations have shown that electrons experience adiabatic acceleration at DFs by betatron and Fermi mechanisms. On the contrary, spacecraft observations and kinetic simulations demonstrated that ions experience nonadiabatic interaction with the DF and can be resonantly accelerated by its electric field

(e.g. Delcourt and Sauvaud, 1994; Delcourt, 2002; Zhou et al., 2010; Greco et al. 2014, 2015, Artemyev et al. 2012, 2015, Ukhorskiy et al. 2013). It was shown, that proton acceleration strongly depends on the velocity of the DF, and the proton energy gain increases with the front amplitude ($B_{Z\_max}$) (Greco et al. 2014). Ukhorskiy et al. 2013 reported that under realistic conditions, the maximum energy gain depends on the dawn-dusk extent of the front. Their simulations demonstrated that the trapped protons can be accelerated up to 100 keV at the DF unbounded in the dawn-dusk direction. The energy gain is limited

because of the nonadiabatic scattering off the equatorial plane due to high magnetic field curvature. Greco et al. 2015 demonstrated that the energy gained by the most energetic fraction of ions scales approximately as the square root of the mass ratio and that ion energization at DFs strongly depends on the initial particle energy.

During the prolonged "secondary" dipolarizations the increases in fluxes of suprathermal electrons, protons and heavy ions are often observed (Nosé et al., 2000; Apatenkov et al., 2007; Asano et al., 2010; Grigorenko et al., 2017; Malykhin et al.,

2018a,b). Malykhin et al 2018b studied the dynamics of fluxes and energy spectra of suprathermal electrons during the prolonged dipolarizations and showed that electrons can be accelerated by betatron mechanism up to ~ 90 keV. On the contrary, the behavior of ion energy spectra indicates the nonadiabatic character of ion acceleration in the course of dipolarizations (e.g. Nosé et al., 2000; Grigorenko et al., 2017; Malykhin et al 2018a). However ion dynamics and acceleration mechanisms operating in the course of prolonged dipolarizations are still poorly understood.

In this work, we study the dynamics and acceleration of ions of different masses (H+, He+, O+) to suprathermal energies in the multiscale magnetic structure of dipolarizations in the near-Earth magnetotail by using Cluster/RAPID observations (Wilken et al., 2001). The structure of the article is as follows. In Sect. 2, we describe the observational data used and show a typical example of the dynamics and spectra of suprathermal H⁺, He⁺, O⁺ ions during dipolarization. In Sect. 3, we present the statistical analysis of the fluxes and spectra of these ion components observed in 17 dipolarization events. The results of our

study are formulated and discussed in Sect. 4.

## 2 Observations

To study the dynamics and acceleration of suprathermal ions we used observations provided by the Research with Adaptive Particle Imaging Detectors (RAPID) spectrometers on board four Clusters spacecraft in the energy range of 40– 1500 keV for protons, and up to 4000 keV for heavier ions (Wilken et al., 2001). The magnetic field observations were taken from the

fluxgate magnetometers (FGMs) (spin- and full resolution (22.4 Hz) modes were used) (Balogh et al., 2001). Ion moments of thermal population were taken from the COmposition DIstribution Function (CODIF) instrument (Réme et al., 2001). CODIF measures proton fluxes in the energy range of 0–40 keV/e. The electric field data were provided by the Electric Field and Wave (EFW) instrument (Gustafsson et al., 2001). If not specially mentioned, we use the geocentric solar magnetospheric (GSM) coordinate system everywhere in the paper.

Figure 1 shows a dipolarization event observed by Cluster-1 (C1) spacecraft on 3 October 2004 between 18:56 – 19:03 UT. The observations from the other Cluster satellites are similar and are not shown. At this time Cluster was located at (-15; 6;





2.5) $R_E$ and it was inside in Plasma Sheet (PS) ($|B_X| < 10$ nT, see Figure 1d). The dipolarization started around 18:57:49 UT. This moment is indicated in Figure 1 by the vertical dotted line. An increase in the earthward ion bulk velocity ($V_X$) was observed simultaneously with the sharp increase in the $B_Z$ field (Figure 1d,f). This indicates the arriving of BBF along with the DF. We will call this DF as the onset-related DF. Since after this front the dipolarization lasted until ~ 19:45 UT and had

the complicated magnetic structure (not shown). Simultaneously with the increase in $V_X$, a sharp increase in the dawn-dusk electric field ($E_Y$) was observed. The $E_Y$ field shown in Figure 1e represents the smoothed full-resolution $E_Y$ data by the 4s - sliding average. After the onset of dipolarization the oscillations of the $V_X$ value and its sign reversal were observed. This indicates the braking and reflection of the fast flow (e.g. Panov et al., 2010). Multiple short $B_Z$– pulses were observed during the $B_Z$ growth (between 18:57:49 and 18:58:30 UT) as well as at the later time when the $B_Z$ field had already reached the large

magnitude.

Figures 1a-c show the time profiles of suprathermal fluxes of $H^+$, $He^+$, $O^+$ ions observed in several energy channels of RAPID instrument. The corresponding average energies are presented in the bottom part of each panel. The increase in fluxes of high-energy ions (~120 - 600 keV for $H^+$, ~ 287 - 1100 keV for $He^+$, ~560 - 1160 keV for $O^+$) started ~ 1 min before the dipolarization onset, and the values of these fluxes remained large during several minutes after the onset. At the same time the ion fluxes in

lower energy range (~45 - 80 keV for $H^+$, ~150 - 200 keV for $He^+$ and ~340 - 450 keV for $O^+$) decreased during the dipolarization. The contrasting dynamics of ion fluxes in different energy ranges caused the variations in energy spectra and the formation of non-monotonic features like flattening and bulges.

To quantify the energy spectra of suprathermal ions we used the value of spectral index $\gamma$. To calculate the $\gamma$ we assume that the ion differential flux ($J_i$) can be described by a power law at least within the energy range corresponding to neighboring

channels of RAPID instrument: $J_i \sim W^{-\gamma}$ , where $W$ is ion kinetic energy (e.g. Øieroset et al., 2002; Imada et al., 2007). We calculated the spectral index $\gamma$ as it was described by Kronberg and Daly (2013):

$\gamma = \ln(J_{i2}/J_{i1})/\ln(E_{eff2}/E_{eff1})$

Here, the $J_{i2}$ and $J_{i1}$ are the differential fluxes of ions in the neighboring energy channels. The effective energies $E_{eff2}$ and $E_{eff1}$ were calculated as the geometric mean between the lowest energies of the neighboring channels.

Figure 2 displays the time profiles of $\gamma$ calculated for three types of ions ($H^+$, $He^+$, $O^+$) for given energy ranges during the interval of interest shown in Figure 1 (panels b-d). We also calculated energy spectra of these ion components at the moments shown by the colored vertical lines in panels (a-d) and presented them in the right part of the Figure. The gray-shaded areas in panels (e-g) indicate the energy ranges at which the spectra have nonmonotonic features (flattening and bulges). Figure 2a shows the time profiles of the $B_Z$ and $B_X$ components for the reference.

Before the dipolarization onset the values of proton's $\gamma$ in energy ranges ~45 – 83 keV (shown by blue line in Figure 2b, $\gamma_{H\_45}$) and ~83 – ~121 keV (shown by red line, $\gamma_{H\_83}$) were similar (~3.0). After the onset, the $\gamma_{H\_45}$ almost did not change, but the $\gamma_{H\_83}$ was decreasing down to ~1.5. The decrease in $\gamma_{H\_83}$ started a few seconds before the onset and lasted until ~18:59:20 UT. The corresponding evolution of proton spectra is displayed in Figure 2e. One can see that the initial spectrum measured at 18:57:30 UT, i.e. before the onset (shown by blue line) has a rather monotonic power-law shape. After the onset the spectrum

flattening in the energy range shaded by grey (~83 – 121 keV) in Figure 2e is observed. This feature reflects the observed significant decrease in $|\gamma_{H\_83}|$. Thus, the proton spectrum becomes more energetic after the dipolarization onset. In the higher energy range (~ 121 – 600 keV) the spectral index $\gamma$ decreased even earlier: ~1 min before the dipolarization onset and it proceeded to decrease after the onset. This dynamics was observed simultaneously with the increases in proton fluxes in the corresponding energy channels (see Figure 1a). The observed negative variations of $\gamma$ indicate proton acceleration near the

onset-related DF as well as within the $B_Z$ pulses observed after the onset.

The dynamics of $He^+$ spectra is more or less similar to the proton's one (see Figure 2c,f). The spectral index $\gamma$ in the lower energy range ~154 – 201 keV (shown by blue line in Figure 2c, $\gamma_{He\_154}$) was decreasing slightly during the entire interval of interest. On the contrary, in the higher energy range ~201 – 508 keV the $\gamma_{He}$ started to decrease ~1 min before the onset of





dipolarization and proceeded to decrease after the onset. The strongest and most prolonged negative variation of $\gamma_{He}$ was observed in 201.3 - 287.3 keV energy range (Figure 2c). Similarly to protons in the energy spectrum of He$^+$ ions the flattening in a finite energy range (~201 – 287 keV) was observed after the dipolarization onset (see the gray-shaded area in Figure 2f).

The dynamics of energy spectra of oxygen ions is different from the dynamics of light ions spectra. In the lower energy range

(~337 – 454 keV, shown by blue line in Figure 2d) the $\gamma_{O\_337}$ experienced a slight increase during the entire interval of interest. In the higher energy range the fluxes of O+ started to grow ~ 30 s before the onset (see Figure 1c). This manifests in a sharp decrease in $\gamma_{O\_563}$ (shown by gold line in Figure 2d). Then, after the onset the $\gamma_{O\_563}$ experienced bipolar variations and its strongest negative variation was observed only by the end of interval of interest around 19:00:20 UT. In the middle energy range (~454 -654 keV, shown by red line in Figure 2d) the decrease in $\gamma_{O\_454}$ also started before the dipolarization onset, but

later than the start of $\gamma_{O\_563}$ decrease. The decrease in $\gamma_{O\_454}$ lasted after the dipolarization onset until ~18:59:20 UT. Thus, there were time delays between variations of $\gamma_{O\_454}$ and $\gamma_{O\_563}$. These features in the dynamics of $\gamma$ at different energy ranges result in observations of bulges in the energy spectra of O$^+$ ions after the dipolarization onset (see Figure 2g).

The dynamics of fluxes and energy spectra of different ion components observed before and after the dipolarization onset indicates the ion acceleration in the limited energy ranges, which occurred at different stages of dipolarization for ions of

different masses. In the next section we present the statistical analysis of these phenomena.

### 3. Statistical studies

In the previous section it was shown, that fluxes of H$^+$, He$^+$ and O$^+$ ions with energies $\geq$ 121 keV, 201 keV and 454 keV respectively increased during the dipolarization, while ion fluxes in lower energy range either decreased or remained unchanged. This behavior caused the negative variations of spectral index $\gamma$. To study statistically the dynamics of ion fluxes

and the $\gamma$ we applied the superposed epoch analysis to 17 dipolarization events from the list published by Grigorenko et al. (2016). Wherein, the O$^+$ fluxes were statistically reliable only in 11 events. The list of events used in our statistical studies is presented in Table 1.

Figure 3 shows the epoch profiles of fluxes (panels a - f) and $\gamma$ (g - k) of H$^+$ ions for 17 events listed in Table 1. Figure 3f and k display the epoch profiles of the $B_Z$ field. For each event the $B_Z$ field was normalized to the maximum value of the $B_Z$ observed

in a given event: $B_Z^*(t) = B_Z(t)/B_{Zmax}$. The proton fluxes observed in a given energy range were also normalized by the similar way. As the epoch time ($t = 0$) we use the dipolarization onset detected in each event. The black lines display the median profiles of fluxes and $B_Z^*(t)$ and grey dashed lines show lower and upper quartiles of the corresponding epoch profiles.

The increase in suprathermal H$^+$ fluxes (~ 83 - 600 keV) started ~1.5 min before the dipolarization onset and it lasted during ~1 min after the onset. During this time the $\gamma$ calculated for this energy range was decreasing (the green-shaded interval in

Figure 3 g-k). However, in the lower energy range (~45 keV) the proton flux hardly changed (see Figure 3a), and the corresponding spectral index $\gamma_{H\_45}$ was almost constant (~3.5, see Figure 3g) during the entire dipolarization intervals. Thus, the decrease in $\gamma$ was observed only in the limited energy range simultaneously with the flux increase in this range. This indicates the proton energization up to the energies $\geq$ 83 keV in the course of dipolarizations. It is worth also noting, that after the dipolarization onset the $\gamma_{H\_83}$ decreased down to zero. This indicates the flattening of proton spectra in the finite energy

range (~83 - 121 keV, see Figure 3h).

Figure 4 presents the epoch profiles of fluxes (panels a - e) and $\gamma$ (f - i) of He$^+$ ions for 17 events from Table 1. The format of the Figure is the same as in Figure 3.The dynamics of He$^+$ fluxes is similar to the dynamics of H$^+$ fluxes. The monotonic increase in helium fluxes (~201 – 508 keV) started ~1.5 min before the dipolarization onset and lasted during ~1 min after the onset (see Figure 4 b - d). Conversely the decrease in spectral index $\gamma$ in ~ 154 – 508 keV although started ~1 min before the

dipolarization onset but was observed longer: during ~3 min after the onset (see green-shaded interval in Figure 4 f - i). Thus, after the dipolarization onset the acceleration of He$^+$ lasted longer than the proton's acceleration.





Figure 5 shows the epoch profiles of fluxes (panels a - e) and the $\gamma$ (f - i) of $O^+$ ions observed in 11 dipolarization events from Table 1. The increase in fluxes of $O^+$ ions (~ 337 – 563 keV) was observed at the similar time scale as the increase in energetic $H^+$ and $He^+$ fluxes (see green-shaded interval in Figure 5 a-e). However, the monotonic decrease in $\gamma$ was observed only in ~454 – 564 keV energy range. It started just before the dipolarization onset and lasted during ~ 5 min after the onset (see green-shaded interval in Figure 5h). In the lower energy range the $\gamma$ experienced bipolar variations which started ~50s before the onset and were observed during ~ 7 min after the onset. The signatures of flattening of energy spectrum ($\gamma \sim 0$) were observed in energy range of ~337 – 454 keV after the onset (Figure 5g). It is worth noting that in the energy spectra of oxygen ions the negative variations of $\gamma$ were much stronger ($\Delta\gamma_{O+} \sim -5 – -3$) than the ones detected in the spectra of light ions ($\Delta\gamma_{H+} \sim -2$, $\Delta\gamma_{He+} \sim -2.5$).

Indeed, in Figure 6 we present the distribution of probability to observe the given values of $\Delta\gamma$ for each ion component and for each energy range calculated for the dipolarization events from our data base. The $\Delta\gamma$ values were calculated in the following way. For each dipolarization event we determine the maximum value of spectral index observed around dipolarization onset ($\gamma_0$), the minimum value $\gamma_1$ observed after the onset, and the maximum value $\gamma_2$ observed during the relaxation of $\gamma$ by the end of the dipolarization. Then, for each event we calculate the negative variation $\Delta\gamma_-$ as $\gamma_1 - \gamma_0$, and the positive variation $\Delta\gamma_+$ as $\gamma_2 - \gamma_1$, and select the strongest one from them. Thus, in Figure 6 we show the probability to observe a given value of either positive or negative $\gamma$ variations observed in each dipolarization event. One can see that the $\Delta\gamma_{H+}$ and $\Delta\gamma_{He+}$ values are mainly ranged between -5.0 – 0.0, while the majority of $\Delta\gamma_{O+}$ is ranged between -10.0 – -3.0. Since the negative variations of $\gamma$ indicate the ion acceleration one may assume that heavy ions ($O^+$) experience stronger acceleration than light ions ($H^+$ and $He^+$) during dipolarizations.

## 4. Discussion

In this paper, we analyze the dynamics and spectra of fluxes of different ion components ($H^+$, $He^+$, $O^+$) in suprathermal energy range (45–700 keV) during dipolarizations in the near-Earth magnetotail ($X > -17$ $R_E$) according to Cluster/RAPID observations made in 2001–2005. By using superposed epoch analysis we show that the increase in the suprathermal ion fluxes ($H+$, $He+$, $O+$) started ~1 min before the onset of dipolarizations, and the fluxes continued to grow during ~1 min after the onset. The fact that the fluxes of different ion components behave in a similar way suggests that their dynamics can be related to the contraction of magnetic flux tubes during dipolarization.

Our observations demonstrate that the spectral index $\gamma$ started to decrease almost simultaneously with the flux increase. But, unlike the behavior of ion fluxes, the decrease in $\gamma$ was observed during different time intervals for different types of ions. Namely, for protons, the decrease in $\gamma$ was observed during ~ 1 min after the onset, while for $He^+$ and $O^+$ ions the $\gamma$ was decreasing during ~ 3 min and ~ 5 min after the onset respectively.

The decrease in $\gamma$ indicates the presence of non-adiabatic acceleration of different ion components during dipolarization. Indeed if ion acceleration were adiabatic the spectral index should be almost constant (e.g. Pan et al., 2012). However, in all dipolarization events from our data base we observed strong negative variations of $\gamma$ in the limited energy range for all ion components. For $O^+$ ions these variations were in ~ 2.5 times larger than for light ions. The amplitude of the negative variations of $\gamma$ and the duration of $\gamma$ decrease demonstrate that oxygen ions experience stronger nonadiabatic acceleration than light ions ($H^+$ and $He^+$). Other important spectral features in favor of nonadiabatic acceleration mechanism are flattening of ion spectra and the formation of bulges in a finite energy range.

Theoretical studies and modeling of ion acceleration in a single DF were performed in many papers before (e.g. Ukhorski et. al. 2013, Perri et al., 2009; Greco et al., 2010). It was shown that the efficiency of charged particles acceleration depends not only on the amplitude of electric field associated with the DF, but also on the spatial structure of this field and the particular location of particle arrival into the acceleration region (Artemyev et. al, 2015). Greco et. al. (2014) showed that the DF





propagating earthward has a complex 2D electric field structure, which is defined by the front propagation velocity ($V_f$) and the following spatial scales: magnetic ramp thickness ($l_f$) and the DF width ($L_f$). It was also shown that the acceleration of charged particles also depends on the initial energy and mass of ions (Greco et. al 2015). Thus, heavy ions ($O^+$) should experience more efficient acceleration than light ions. According to their estimations the characteristic energy gain for $H^+$ ions

is ~ 20 - 80 keV, for $He^+$ - ~ 20 - 140 keV and for $O^+$ -~ 40 - 240 keV, in dependence on the DF velocity ranged from 200 to 800 km/s.

Assuming the same acceleration mechanism, we suggest that ions obtain energy by passing the potential drop in the *Y*-direction, in the course of their interaction with the DF. To check this, we estimated the front parameters necessary for such acceleration. By using Minimum Variance Analysis and timing analysis (Sonnerup and Scheible, 1998) we determined the direction and

value of propagation velocity of the DFs in the *(XY)* plane ($V_f$) for each dipolarization event from our data base (see Table 2). Then, by using the $V_f$ value and the amplitude of the $B_Z$ variation ($\Delta B_Z$) at the DFs we estimated roughly the electric field associated with the front: $E = \Delta Bz \cdot V_f$. Since in the course of nonadiabatic interaction with the DF ions obtain energy at spatial scale $\sim 2r_L$ ($r_L$ is the ion gyroradius), one can estimate roughly the minimum ion energy gain as $\Delta W = 2E \cdot r_L$.

It is worth noting that in each dipolarization event from our data base we observed the decrease in ion fluxes in the lowest

RAPID channels along with the increase in fluxes in higher energy channels. This effect was observed for all types of ions (see Figure 1). It can be due to the nonadiabatic acceleration of ion populations with initial energies corresponding to the lowest RAPID channels. This results in replenishment of more energetic ion population and the corresponding flux increase in higher energy channels. Based on these observations we used for $r_L$ calculation the ion energy corresponding to the lowest RAPID channels for each type of ions. Since in the course of ion nonadiabatic interaction with the DF the $r_L$ changes both due

to the increase in |*B*| at the front and due to the increase in ion energy, then for the rough estimate of $r_L$ we use the average value of the |*B*|: $<B> = (B_{min} + B_{max})/2$, where $B_{min}$ is the minimum value of the magnetic field observed at the dip region before the DF (e.g. Shiokawa et al., 2005) and $B_{max}$ is the maximum value of the magnetic field corresponding to the DF.

Figure 7 shows the probability distribution of the $\Delta W$ of each ion component estimated for all events from our data base. The normal mean value for the distributions are ~ 24 keV for $H^+$, ~ 89 keV for $He^+$ and ~ 223 keV for $O^+$ ions. The estimated

values of energy gain are enough to transfer some fraction of ion population from the lower part of suprathermal energy distribution to the higher energy range. These results are within the energy ranges obtained by Greco et. al 2015 for the energy gain provided by the nonadiabatic acceleration in the course of ion interaction with a single propagating DF.

However, our analysis of the *γ* dynamics showed that some fraction of light ions can be accelerated to energies more than 600 keV and some fraction of oxygen ions can be accelerated up to ~1.2 MeV. Thus, in some cases the amount of energy gain may

exceed both the theoretical and our own estimations of $\Delta W$. It is worth noting that these estimations were obtained for a single propagating DF. However, dipolarizations analyzed in our study represent long lasting complicated events, which consist of multiple DFs with different spatial scales. Ions can experience multistage energy gain in the course of their nonadiabatic interaction with multiple DFs.

### 5. Conclusion

In this paper we studied the dynamics of fluxes and spectra of suprathermal ions of different masses during dipolarizations in the near-Earth geomagnetic tail ($X > -17$ $R_E$) according to Cluster/RAPID observations made in 2001–2005. Below we summarize our main results:

1. During dipolarizations in the near-Earth magnetotail the increase in fluxes of different ion components were observed in the following energy ranges: ~ 120 - 600 keV for $H^+$, ~ 287 - 1100 keV for $He^+$, and ~ 560 - 1160 keV for $O^+$ ions. These increases

started ~ 1 min before the onset of dipolarizations and lasted during ~ 1 min after the onset. Simultaneously with the flux



increase in higher energy channels the decrease in fluxes in lower energies channels were detected. This indicates the presence of nonadiabatic effects in the dynamics of the suprathermal ion fluxes.

2. Simultaneously with the start of fluxes increase the decrease in the value of spectral index $\gamma$ was observed. The duration and amplitude of negative variations of $\gamma$ depend on the ion mass. In each dipolarization event from our data base the acceleration of heavy ions was observed after the onset during the longer time interval than the acceleration of light ions.

3. The largest amplitude of negative variations in $\gamma$ was detected for heavy ions ($O^+$). This demonstrates that more efficient acceleration is observed for heavy ions than for light ions ($H^+$, $He^+$).

4. The strong negative variations of $\gamma$ were observed in finite energy ranges for all ion components. This indicates the possibility of non-adiabatic resonant acceleration of ions in the course of their interaction with multiple DFs during dipolarizations.

*Data availability.* In this paper we only used open-access data. The Cluster data were downloaded from the Cluster Science Archive version 1.2.1 at http://www.cosmos.esa.int/web/csa (last access: 5 March 2018). To obtain the data, one should start the CSA GRAPHICAL USER INTERFACE, and then to download the data, the particular instrument and time interval should be selected.

**Acknowledgements**

The work of A. Yu. Malykhin and E. E. Grigorenko was supported by Russian Science Foundation (grant # 18-47-05001). The work of E. A. Kronberg and L.V. Kozak was supported by the Volkswagen Foundation (grant Az 90 312). E. A. Kronberg and P. W. Daly acknowledge the "Deutsches Zentrum für Luft und Raumfahrt (DLR)" for the support of the RAPID instrument under grant 50 OC 1602.





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



## Captions

**Figure 1.** An overview of the dipolarization event observed on 3 October 2004 by Cluster-1. Panels (a) - (c) show the time profiles of fluxes of suprathermal $H^+$, $He^+$ and $O^+$ ions respectively measured by RAPID instrument in the energy ranges indicated at the corresponding panel. Panel (d) shows time profiles of the $B_X$ and $B_Z$ components of the magnetic field. Panels (e) and (f) display the dawn-dusk electric field ($E_Y$) and $X$-component of ion bulk velocity ($V_X$). The onset of dipolarization is indicated by vertical dotted line.

**Figure 2.** Variations of spectral index $\gamma$ calculated for $H^+$ (b), $He^+$ (c) and $O^+$ (d) ion components in the energy ranges indicated at the corresponding panels. Panel (a) shows the time profiles of $B_X$ and $B_Z$ field for reference. Vertical dashed colored lines indicate the time moments at which energy spectra of $H^+$, $He^+$ and $O^+$ ions are plotted in panels (e) – (d) respectively. The grey-shaded areas at panels (e) - (d) display the energy ranges at which the peculiarities (flattening or bulges) are observed in the energy spectra.

**Figure 3.** Panels (a) – (e) show the epoch profiles of $H^+$ ion fluxes with energies indicated at the corresponding panel. Panels (g) – (j) display the epoch profiles of $\gamma$ calculated in the given energy ranges indicated at the corresponding panel. Panels (f) and (k) show the epoch profile of the normalized $B_Z$ magnetic field (see explanation in the text). Vertical dotted line marks the epoch time $t = 0$ which corresponds to dipolarization onset in each event from our data base. The green-shaded area at panels (g) – (k) displays the time interval of $\gamma$ decrease. The black lines display the median profiles of fluxes and $B_Z^*(t)$ and grey dashed lines show lower and upper quartiles of the corresponding epoch profiles.

**Figure 4.** The epoch profiles for $He^+$ ions. The format of the Figure is similar to the format of Figure 3.

**Figure 5.** The epoch profiles for $O^+$ ions. The format of the Figure is similar to the format of Figure 3.

**Figure 6.** Distribution of probability to observe the given values of $\gamma$ variations (see explanation in the text) for $H^+$, $He^+$ and $O^+$ ion components. For each ion component the energy ranges used for $\gamma$ calculation are indicated by colors according to a legend shown in the corresponding panel.

**Figure 7.** Histograms of probability to observe the given values of $H^+$, $He^+$ and $O^+$ ion energy gain in the course of nonadiabatic acceleration at the onset-related DF (see explanation in the text).



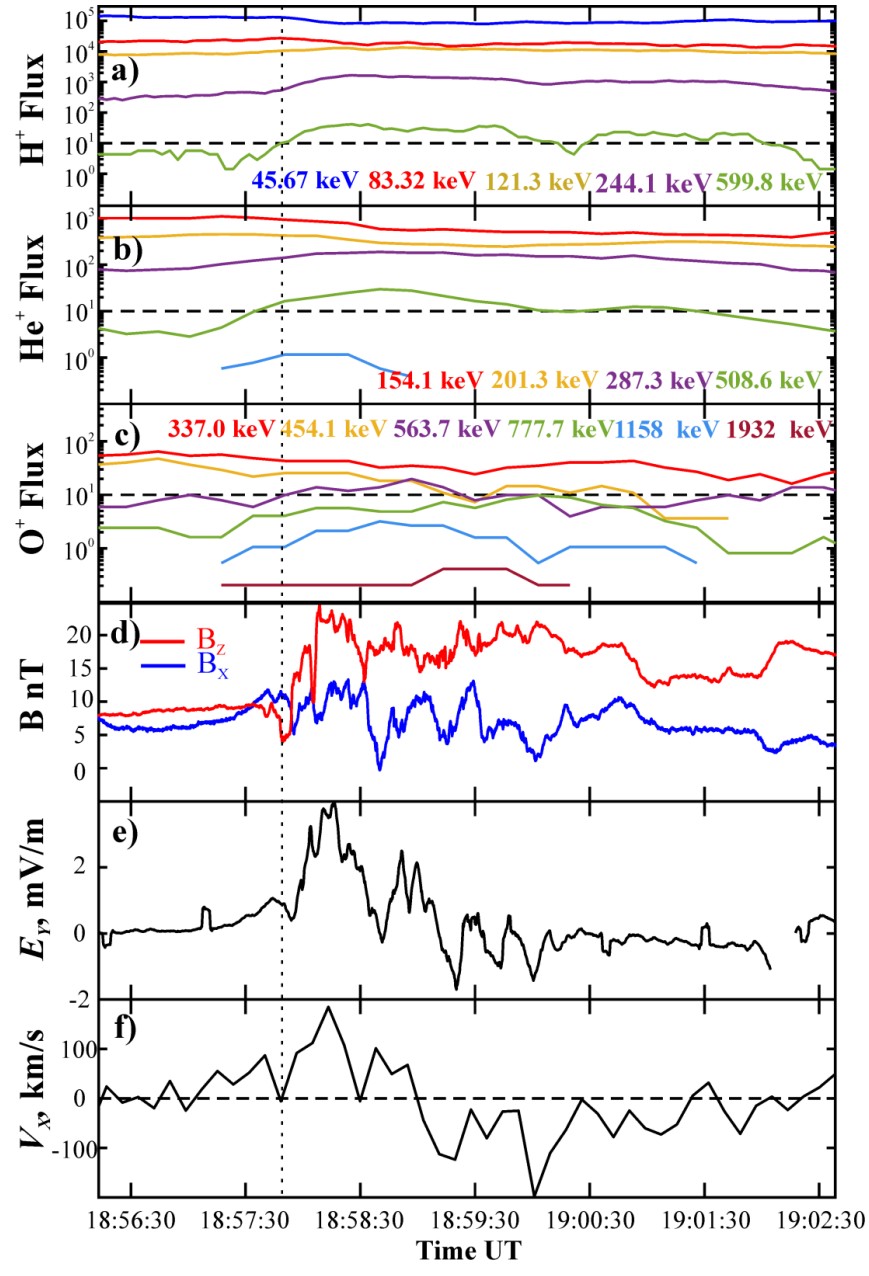

Figure 1.

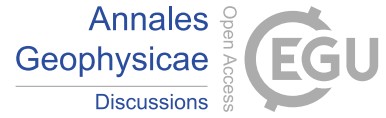

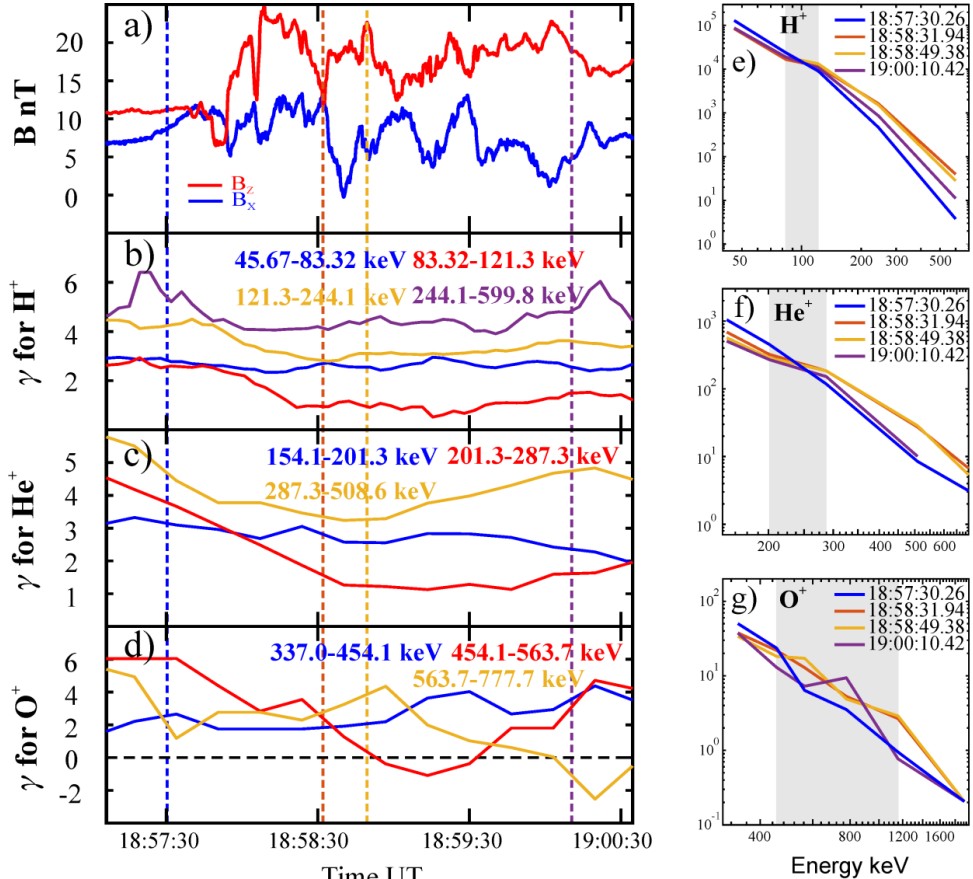

**Figure 2.**





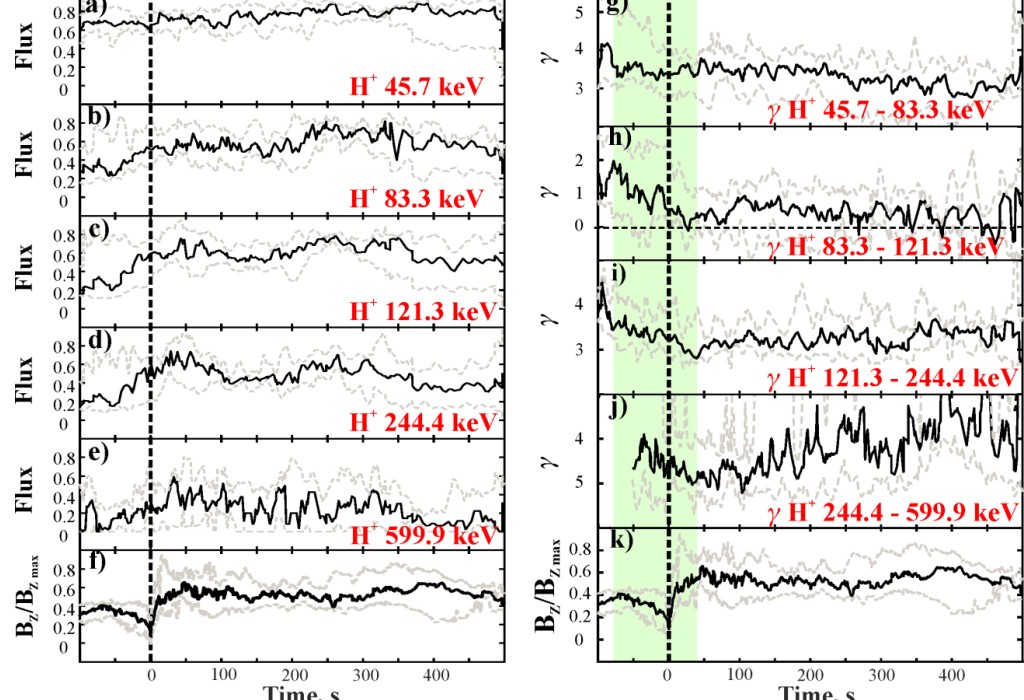

**Figure 3.**





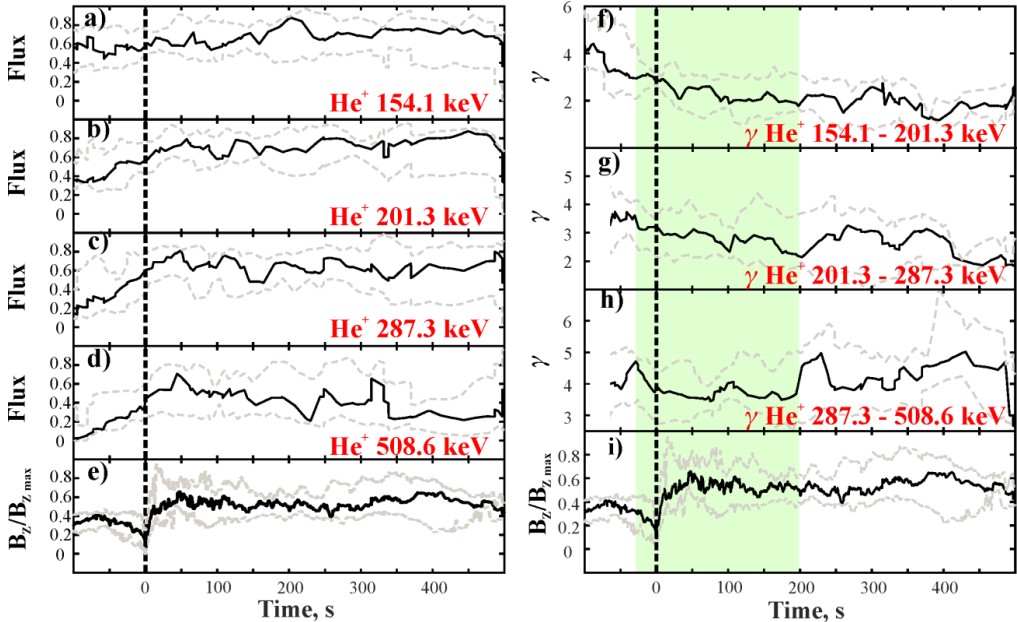

**Figure 4.**





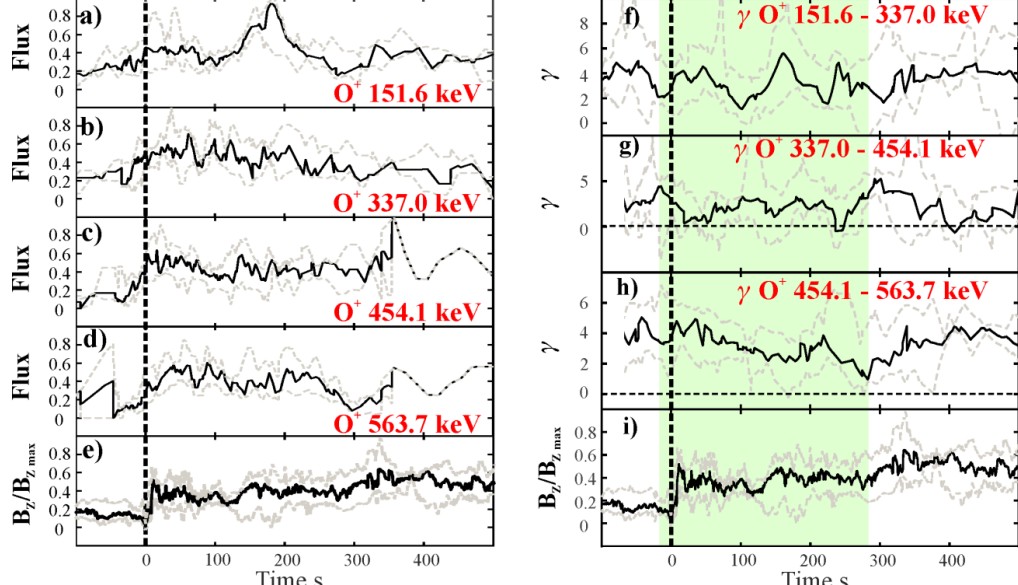

**Figure 5.**





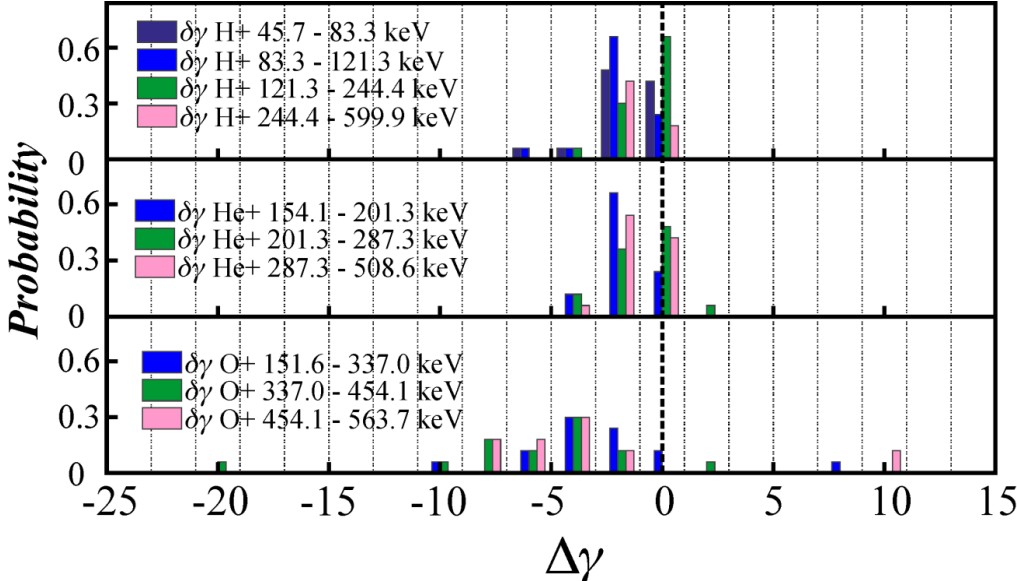

**Figure 6.**




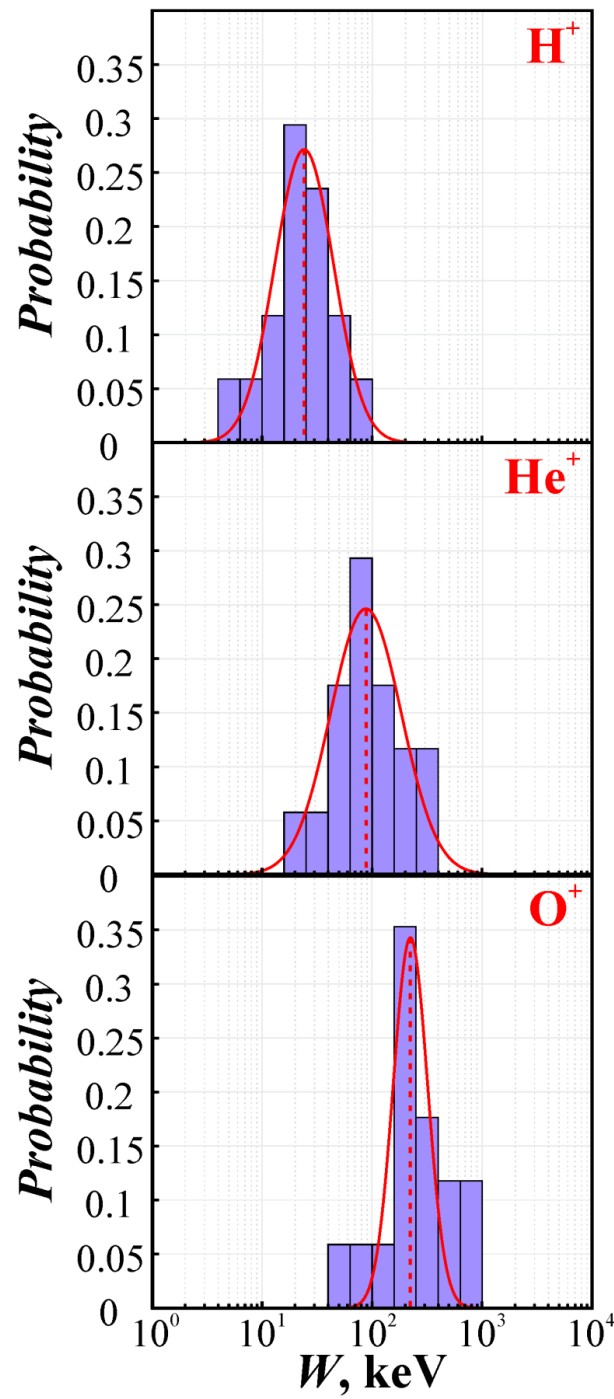

**Figure 7.**



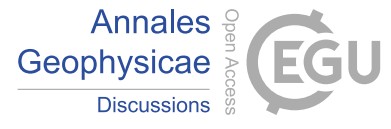

| Date | H+ / He+ | O+ |
|---|---|---|
| 2001-08-12 18:17 | + | |
| 2001-09-15 00:43 | + | + |
| 2001-09-15 01:02 | + | |
| 2002-10-26 07:27 | + | |
| 2003-07-25 07:05 | + | + |
| 2003-07-25 07:21 | + | + |
| 2003-07-27 13:45 | + | + |
| 2003-07-29 18:36 | + | + |
| 2003-08-01 06:32 | + | |
| 2003-09-24 15:10 | + | |
| 2003-09-24 16:07 | + | + |
| 2004-10-03 18:57 | + | |
| 2004-10-11 01:35 | + | + |
| 2004-10-13 05:49 | + | + |
| 2004-10-13 06:55 | + | + |
| 2005-08-09 18:29 | + | + |
| 2005-09-28 17:13 | + | + |

Table 1. The list of time moments of the dipolarization onset observed in the events from our data base. "+" marks those events, in which the reliable fluxes of $H^+$, $He^+$ and $O^+$ ions were detected by RAPID instrument.



| Time UT | $V_{f,}$ (km/s) | $B_{Z\_max}$ | $\Delta B_Z$ | $|B|_{dip}$ | $|B|_{max}$ |
|---|---|---|---|---|---|
| 2001-08-12 18:17 | 174 | 14.5 | 15.6 | 8.8 | 20.0 |
| 2001-09-15 00:43 | 294 | 17.2 | 16.3 | 7.4 | 23.3 |
| 2001-09-15 01:02 | 320 | 19.9 | 16.1 | 16.1 | 21.0 |
| 2003-07-25 07:05 | 180 | 12.1 | 10.6 | 22.3 | 24.4 |
| 2003-07-25 07:06 | 347 | 13.8 | 11.3 | 24.7 | 25.8 |
| 2003-07-25 07:21 | 270 | 19.3 | 19.5 | 14.8 | 23.2 |
| 2003-07-25 07:29 | 227 | 20.1 | 10.9 | 16.2 | 24.5 |
| 2003-07-27 13:45 | 274 | 14.1 | 10.3 | 11.7 | 19.3 |
| 2003-07-29 18:36 | 232 | 21.4 | 13.9 | 13.3 | 24.3 |
| 2003-07-29 18:46 | 328 | 22.1 | 11.1 | 15.4 | 26.7 |
| 2003-08-01 06:32 | 300 | 16.1 | 16.6 | 4.8 | 19.2 |
| 2003-08-01 06:35 | 378 | 14.5 | 15.4 | 5.7 | 17.3 |
| 2003-09-24 15:10 | 646 | 15.8 | 11.8 | 4.6 | 17.4 |
| 2003-09-24 15:11 | 714 | 17.2 | 10.1 | 8.4 | 19.3 |
| 2003-09-24 16:07 | 279 | 25.5 | 29.6 | 18.5 | 26.7 |
| 2003-09-24 16:10 | 341 | 27.1 | 27.2 | 20.1 | 28.2 |
| 2004-10-03 18:57 | 352 | 26.5 | 21.5 | 7.6 | 29.9 |

**Table 2.** Parameters of the DFs observed in dipolarization events from our data base. Namely, the values of propagation velocity of the DFs ($V_f$) in the ($XY$) plane, maximum values of the $B_Z$ field ($B_{Z\_max}$), the amplitude of the $B_Z$ field variation ($\Delta B_Z$) associated with the DFs, magnetic field value in the dip region observed before the DFs ($|B|_{dip}$) and the maximum values of the magnetic field at the DFs ($|B|_{max}$).