# Peer review of "Acceleration of protons and heavy ions to suprathermal energies during dipolarizations in the near-Earth magnetotail"

_Annales Geophysicae, 2019_

## Referee Comment (RC1) · Anonymous Referee #1 · 28 Apr 2019

Report on the manuscript "Acceleration of protons and heavy ions to suprathermal energies during dipolarizations in the near-Earth magnetotail" by Malykhin et al.

This manuscript reports a detailed study of ion acceleration in the near-Earth magnetotail, considering both protons, singly charged helium and singly charged oxygen. Ion acceleration is studied during complex dipolarization events with data obtained by Cluster spacecraft. Both the flux increases and the change in spectral index are considered in order to asses the acceleration of the different ion species. It is found that the flux increases start simultaneously for all species, but they last longer for heavier ions. Also, the change of spectral index, i.e. a decrease in the exponent \gamma

which implies a hardening of the spectrum, is larger for helium and even larger for oxygen. It is also found that oxygen ions are those which reach the highest energies in the considered dataset.

The manuscript is clear and well written, the figures support the conclusions of the text, and I read it with interest. It is appropriate for publication on Ann. Geophysicae, but I suggest the authors to strengthen the physical interpretation by considering the following specific comments.

1. What is the reason for the longer duration of the decrease of $\gamma$ with growing mass? Time of flight?

2. Is heavy ion acceleration mass proportional? From Figure 7 one would say almost, but not exactly. Can you discuss this proportionality? Knowing this would help to contrain the acceleration mechanisms.

3. Have you considered the possibility that multiply charged oxygen ions could influence the measurements?

4. Page 3, line 23: I wonder whether a minus sign should be somewhere in the expression of $\gamma$.

5. Page 4, line 40: "but was observed longer" perhaps would be better "was observed for a longer period"

6. Page 5, lines 25-26: the sentence "The fact that the fluxes of different ion components behave in a similar way suggests that their dynamics can be related to the contraction of magnetic flux tubes during dipolarization" seems to be not related to the rest. Do you mean betatron acceleration? The physical meaning is not clear, and apparently in contrast with the indications of nonadiabatic ion acceleration.

---

## Referee Comment (RC2) · Anonymous Referee #2 · 7 May 2019

This paper studies the dynamics of suprathermal ions of different masses (H+, He+, O+) during prolonged dipolarizations in the near-Earth magnetotail. The acceleration mechanisms signatures are analyzed and their effects on the particle fluxes and spectra of different ion species assessed using a superposed epoch analysis method. The work addresses an important issue, since ion acceleration is one of the main phenomena related to magnetotail dynamics. The analysis is careful and the paper is well written. The paper should be accepted for publication after a minor revision.

Specific Comment:

Discussion, page 6, lines 28 – 33: I suggest to add some discussion on why, in some

cases, the amount of energy gain may exceed both the theoretical and the authors' estimations of $\Delta$W.

---

## Author Comment (AC2) · 27 May 2019

Dear commentator, thanks a lot for your comments and attention to our manuscript. Follow we take our answer: Discussion, page 6, lines 28 – 33: I suggest to add some discussion on why, in some cases, the amount of energy gain may exceed both the theoretical and the authors' estimations of $\Delta W$

-Thank you for this comment. We were not clear enough. In new version we add some discussion in page 6, lines 28 – 33 :

"However, our analysis of the $\gamma$ dynamics showed that some fraction of light ions can

be accelerated to energies more than 600 keV and some fraction of oxygen ions can be accelerated up to ∼1.2 MeV. Thus, in some cases the amount of energy gain may exceed both the theoretical and our own estimations of $\Delta W$. It is worth noting that these estimations were obtained for a single propagating DF. However, dipolarizations analyzed in our study represent long lasting complicated events, which consist of multiple DFs with different spatial scales. In the course of interactions with such multiscale magnetic structures ions can experience multistage energy gain. We may suggest that ions are accelerated due to their subsequent nonadiabatic interactions with the system of multiple DFs and, thus, their resulting energy gain can exceed the energy gain estimated for the interaction with a single DF. Verification of this assumption requires simulation of ion dynamics in complicated multiscale dipolarizations."
* * *

---

## Author Response (AR1)

Response to Reviewer 1

Dear Reviewer,

Thanks a lot for your comments which help us to improve our manuscript. Below we present our answers:

**1. What is the reason for the longer duration of the decrease of \gamma with growing mass? Time of flight?**

Yes. We assume that a longer duration of decrease in gamma is due to the time of flight. The heavy ion velocity is lower than light ion velocity for the same energies. Also, the scales of region of ion interaction with the magnetic structures depend on the ion gyroradius. This assumption is more or less in agreement with our observations.

**2. Is heavy ion acceleration mass proportional? From Figure 7 one would say almost, but not exactly. Can you discuss this proportionality? Knowing this would help to contrain the acceleration mechanisms.**

Thanks for the good question. In Figure 7 we presented theoretical estimation of possible energy gain for different ion masses. This estimation suggests mass-dependent mechanism since the energy gain is proportional to particle gyroradius ($\Delta W = 2E \cdot r_L$). We plotted these histograms to show that the estimated energy gains are of the order of the observed ion energies. Some indications on the presence of mass-dependent acceleration mechanism can be seen in Figures 3 – 5 (right panels). It is seen that the duration of decrease in the value of spectral index γ is longer for heavy ions than for light ions. Also the amplitude of negative variations of γ depends on the ion mass. We discuss this in Page 5, line 27-37.

**3. Have you considered the possibility that multiply charged oxygen ions could influence the measurements?**

Thank you very much for this good question. We do not consider the possibility of RAPID contamination by multiply charged oxygen ions. This ion component is very scarce in the solar wind (<1%), and it's concentration in the Earth magnetotail should be very small and does not influence on our results.

**4. Page 3, line 23: I wonder whether a minus sign should be somewhere in the expression of \gamma.**

Yes! The minus sign must be before the expression. Thank you for finding this typo. We corrected this. But this typo does not influence on our results.

**5. Page 4, line 40: "but was observed longer" perhaps would be better "was observed for a longer period"**

Thanks, we accept your comment.

**6. Page 5, lines 25-26: the sentence "The fact that the fluxes of different ion components behave in a similar way suggests that their dynamics can be related to the contraction of magnetic flux tubes during dipolarization" seems to be not related to the rest. Do you mean betatron acceleration? The**

**physical meaning is not clear, and apparently in contrast with the indications of nonadiabatic ion acceleration.**

Thank you for this comment. We were not clear enough. We meant that the simultaneous start of the flux growth and the similar interval of flux increase observed for different ion components are related to the contraction of magnetic flux tubes. The contraction of magnetic flux tubes can lead to the increase in ion density, but this is not related to the ion acceleration. But, the amplitudes of flux increases are different for different ion energies and masses. This results in the observed variations of gamma indicating on non-adiabatic particle acceleration. Nonadiabatic ion acceleration is related with the ion interactions with DFs. This process can occur simultaneously with the contraction of magnetic flux tubes during dipolarization. We corrected our sentences as follows:

"The fact that the fluxes of different ion components started to increase simultaneously and the duration of the flux growth is similar for ion of different masses can be related to the contraction of magnetic flux tubes during dipolarization. The contraction of magnetic flux tubes can lead to the increase in ion density, which, in turn, leads to the flux increase. However, our observations demonstrate that the spectral index γ started to decrease almost simultaneously with the flux increase. But, unlike the behavior of ion fluxes, the decrease in γ was observed during different time intervals for different types of ions…"

Response to Reviewer 2

Dear Reviewer,

Thanks a lot for your comments which help us to improve our manuscript. Below we present our answers:

**Discussion, page 6, lines 28 – 33: I suggest to add some discussion on why, in some cases, the amount of energy gain may exceed both the theoretical and the authors' estimations of ΔW**

Thank you for this comment. We were not clear enough. In new version we add some discussion in page 6, lines 28 – 33 :

*"However, our analysis of the γ dynamics showed that some fraction of light ions can be accelerated to energies more than 600 keV and some fraction of oxygen ions can be accelerated up to ~1.2 MeV. Thus, in some cases the amount of energy gain may exceed both the theoretical and our own estimations of ΔW. It is worth noting that these estimations were obtained for a single propagating DF. However, dipolarizations analyzed in our study represent long lasting complicated events, which consist of multiple DFs with different spatial scales. In the course of interactions with such multiscale magnetic structures ions can experience multistage energy gain. We may suggest that ions are accelerated due to their subsequent nonadiabatic interactions with the system of multiple DFs and, thus, their resulting energy gain can exceed the energy gain estimated for the interaction with a single DF. Verification of this assumption requires simulation of ion dynamics in complicated multiscale dipolarizations."*